# Exoproducts of the Most Common *Achromobacter* Species in Cystic Fibrosis Evoke Similar Inflammatory Responses *In Vitro*

Cecilia Sahl,[a,b] Maria Baumgarten,[a] Oonagh Shannon,[a] Lisa I. Påhlman[a,b,c]

aDepartment of Clinical Sciences Lund, Division of Infection Medicine, Lund University, Lund, Sweden
bWallenberg Centre for Molecular Medicine, Lund University, Lund, Sweden
cDivision of Infectious Diseases, Skåne University Hospital Lund, Lund, Sweden

**ABSTRACT** *Achromobacter* is a genus of Gram-negative rods, which can cause persistent airway infections in people with cystic fibrosis (CF). The knowledge about virulence and clinical implications of *Achromobacter* is still limited, and it is not fully established whether *Achromobacter* infections contribute to disease progression or if it is a marker of poor lung function. The most commonly reported *Achromobacter* species in CF is *A. xylosoxidans*. While other *Achromobacter* spp. are also identified in CF airways, the currently used Matrix-Assisted Laser Desorption/Ionization Time Of Flight Mass Spectrometry (MALDI-TOF MS) method in routine diagnostics cannot distinguish between species. Differences in virulence between *Achromobacter* species have consequently not been well studied. In this study, we compare phenotypes and proinflammatory properties of *A. xylosoxidans*, *A. dolens*, *A. insuavis*, and *A. ruhlandii* using *in vitro* models. Bacterial supernatants were used to stimulate CF bronchial epithelial cells and whole blood from healthy individuals. Supernatants from the well-characterized CF-pathogen *Pseudomonas aeruginosa* were included for comparison. Inflammatory mediators were analyzed with ELISA and leukocyte activation was assessed using flow cytometry. The four *Achromobacter* species differed in morphology seen in scanning electron microscopy (SEM), but there were no observed differences in swimming motility or biofilm formation. Exoproducts from all *Achromobacter* species except *A. insuavis* caused significant IL-6 and IL-8 secretion from CF lung epithelium. The cytokine release was equivalent or stronger than the response induced by *P. aeruginosa*. All *Achromobacter* species activated neutrophils and monocytes *ex vivo* in a lipopolysaccharide (LPS)-independent manner. Our results indicate that exoproducts of the four included *Achromobacter* species do not differ consistently in causing inflammatory responses, but they are equally or even more capable of inducing inflammation compared with the classical CF pathogen *P. aeruginosa*.

**IMPORTANCE** *Achromobacter xylosoxidans* is an emerging pathogen among people with cystic fibrosis (CF). Current routine diagnostic methods are often unable to distinguish *A. xylosoxidans* from other *Achromobacter* species, and the clinical relevance of different species is still unknown. In this work, we show that four different *Achromobacter* species relevant to CF evoke similar inflammatory responses from airway epithelium and leukocytes *in vitro*, but they are all equally or even more proinflammatory compared to the classic CF-pathogen *Pseudomonas aeruginosa*. The results suggest that *Achromobacter* species are important airway pathogens in CF, and that all *Achromobacter* species are relevant to treat.

**KEYWORDS** *Achromobacter xylosoxidans*, *Achromobacter insuavis*, *Achromobacter dolens*, *Achromobacter ruhlandii*, *Pseudomonas aeruginosa*, cystic fibrosis, *in vitro* inflammatory responses

Address correspondence to Lisa I. Påhlman, lisa.pahlman@med.lu.se.

The authors declare no conflict of interest.

In cystic fibrosis (CF), inherited mutations in the cystic fibrosis transmembrane regulator (CFTR) chloride channel cause accumulation of thick mucus in the airways and gastrointestinal tract. A hallmark symptom of the disease is progressive lung function decline, as microbial pathogens thrive in the mucus. Persistent microbial airway infection and the associated immune responses are the main drivers of lung damage and disease progression (1). The inflammation contributing to CF lung damage is strongly driven by neutrophils (2). Monocytes and alveolar macrophages have also been implicated, as their respective cell counts and chemokine activity have been reported to be increased in CF lungs (3, 4). Monocytes are the precursor cells of macrophages, which have a hyperinflammatory phenotype in CF (5, 6). Treatment of persistent lung infections in CF patients will remain important even as CFTR modulating therapies become more commonplace, as this treatment strategy appears to have a limited effect on bacterial clearance in patients with low microbial diversity (7).

*Achromobacter* is a genus of Gram-negative rods of environmental origin, which are considered emerging pathogens in CF (8). It is currently controversial whether *Achromobacter* infections are causative of harmful lung inflammation, or if they are opportunistic colonizers of lungs that have been damaged by other causes. Different studies have come to different conclusions, likely due to the small number of patients and different study endpoints. Whereas some studies did not find a significant decline in forced expiratory volume per second ($FEV_1$) in colonized patients (9–11), others have shown a decline in lung function after infection (12–15), as well as increased frequencies of hospitalizations, exacerbations, and need for antibiotic treatment (9, 16). Recently, a large study on people with CF in 40 European countries demonstrated that *Achromobacter* infection was associated with disease severity similar to infection with *P. aeruginosa* (17).

The most common *Achromobacter* species in CF is *A. xylosoxidans* (18, 19), but other species are also reported. During diagnostic culturing and identification, various *Achromobacter* species are frequently identified as *A. xylosoxidans* due to technical limitations (20), and by some older methods such as morphological and biochemical phenotyping even as *P. aeruginosa* or *Burkholderia cepacia* complex (21). Whether or not it is clinically relevant to better discriminate between different *Achromobacter* species is not fully understood, as the virulence of both *A. xylosoxidans* and other *Achromobacter* species is poorly characterized (22).

In this study, we investigated the proinflammatory properties of four different *Achromobacter* species relevant to CF airway infections, and compared their inflammatory responses to the well-characterized CF pathogen *P. aeruginosa*.

## RESULTS

**Species distribution among clinical Achromobacter isolates.** A total of 21 clinical isolates identified as *A. xylosoxidans* by MALDI-TOF were subjected to *nrdA* sequencing to confirm the species identity. All isolates were obtained from individual patients. Sequencing of the *nrdA* locus revealed that six of the 21 isolates (29%) were in fact non-*xylosoxidans* species. These isolates belonged to the species *A. insuavis* ($n = 3$), *A. dolens* ($n = 2$), and *A. ruhlandii* ($n = 1$).

**Phenotypical and morphological differences between *Achromobacter* species.** In order to compare the different *Achromobacter* species, a total of five isolates of *A. xylosoxidans*, five *A. insuavis*, four *A. dolens*, and three *A. ruhlandii* were included in the study. This collection included both type strains, on which the description of the species is based, and clinical isolates of each species. In addition, four *P. aeruginosa* isolates were included for comparison (see Table 1). All bacterial species were first characterized phenotypically with regard to swimming motility and biofilm formation capacity (Fig. 1). Swimming motility on agar varied greatly among strains within each species, but there was no statistical difference in motility between species. Biofilm formation was significantly stronger in *P. aeruginosa* compared with *A. xylosoxidans* and *A. dolens*, but there was no significant difference in biofilm formation between the four *Achromobacter* species.

Finally, we performed scanning electron microscopy (SEM) of *A. xylosoxidans*, *A. insuavis*, *A. dolens*, and *A. ruhlandii* type strains to illustrate the morphological differences between

**TABLE 1** List of bacterial isolates used in the study

| Species | Strain | Origin | Type of infection |
| --- | --- | --- | --- |
| *A. xylosoxidans* | CCUG-56438T | Type strain, human ear infection, Japan (CCUG) | NA[a] |
| | 1373 | CF sputum, Lund, Sweden | Sporadic |
| | 1191 | CF sputum, Lund, Sweden | Sporadic |
| | 1362 | CF sputum, Lund, Sweden | Chronic |
| | 1267 | CF sputum, Lund, Sweden | Chronic |
| *A. insuavis* | CCUG-62426T | Type strain, human non-CF sputum, USA (CCUG) | NA |
| | CCUG-62425 | CF sputum, Spain (CCUG) | Unknown |
| | 0880 | CF sputum, Lund, Sweden | Chronic |
| | 0714 | CF sputum, Lund, Sweden | Chronic |
| | 1721 | CF sputum, Lund, Sweden | Intermittent |
| *A. dolens* | CCUG-62421T | Type strain, human non-CF sputum, USA | NA |
| | CCUG-70224 | CF sputum, Gothenburg, Sweden (CCUG) | Unknown |
| | 0753 | CF sputum, Lund, Sweden | Sporadic |
| | 1201 | CF sputum, Lund, Sweden | Chronic |
| *A. ruhlandii* | CCUG-57103T | Type strain, soil sample (CCUG) | NA |
| | CCUG-62287 | CF sputum, Denmark (CCUG) | Unknown |
| | 1017a | CF sputum, Lund, Sweden | Chronic |
| *P. aeruginosa* | PAO1 | Reference strain, wound isolate | NA |
| | PsA3 | CF sputum, Lund, Sweden | Unknown |
| | PsA4 | CF sputum, Lund, Sweden | Unknown |
| | PsA9 | CF sputum, Lund, Sweden | Unknown |

[a]NA, not applicable.

the four species. *A. xylosoxidans* (Fig. 2A) and *A. ruhlandii* (Fig. 2B) appear rod-shaped, whereas *A. insuavis* (Fig. 2C) and *A. dolens* (Fig. 2D) have a filamentous appearance. *A. xylosoxidans* appears to have a larger amount of flagellae than the other species.

**Exoproducts from *Achromobacter* spp. induce an inflammatory response from airway epithelial cells.** We next investigated the ability of exoproducts from the four different *Achromobacter* species to evoke an inflammatory host response *in vitro*. Supernatants from *P. aeruginosa* were included for comparison. CF bronchial epithelial cells were stimulated with supernatants from bacterial overnight cultures, and cytokine release was then quantified in the cell medium. After incubation, cells were visually inspected to ensure viability. No differences were observed between cells exposed to bacterial supernatants and the negative control (Fig. S2), suggesting no cytotoxic effects. Supernatants from all *Achromobacter* species except *A. insuavis* induced a significant IL-6 and IL-8 response from CF lung epithelium (Fig. 3A and B and Fig. S3A and B). *A. xylosoxidans* gave rise to the highest mean cytokine response of all tested *Achromobacter* spp, but this was only statistically significant compared to *A. insuavis* (Fig. 3A and B).

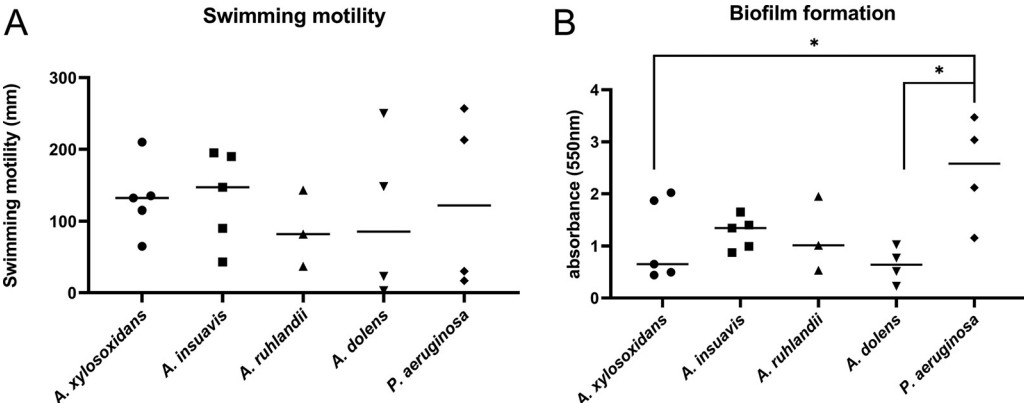

**FIG 1** Phenotypic characterization of *Achromobacter* spp. And *P. aeruginosa*. No difference was observed between species in the swimming motility assay (A). Two of the *Achromobacter* species, *A. xylosoxidans* and *A. dolens*, formed significantly less biofilm than *P. aeruginosa*, with no significant differences observed between the Achromobacter species (B) (Mann-Whitney U test, *, $P < 0.05$). Symbols represent the average value of 3 replicate experiments of each bacterial isolate. Bars illustrate the mean.

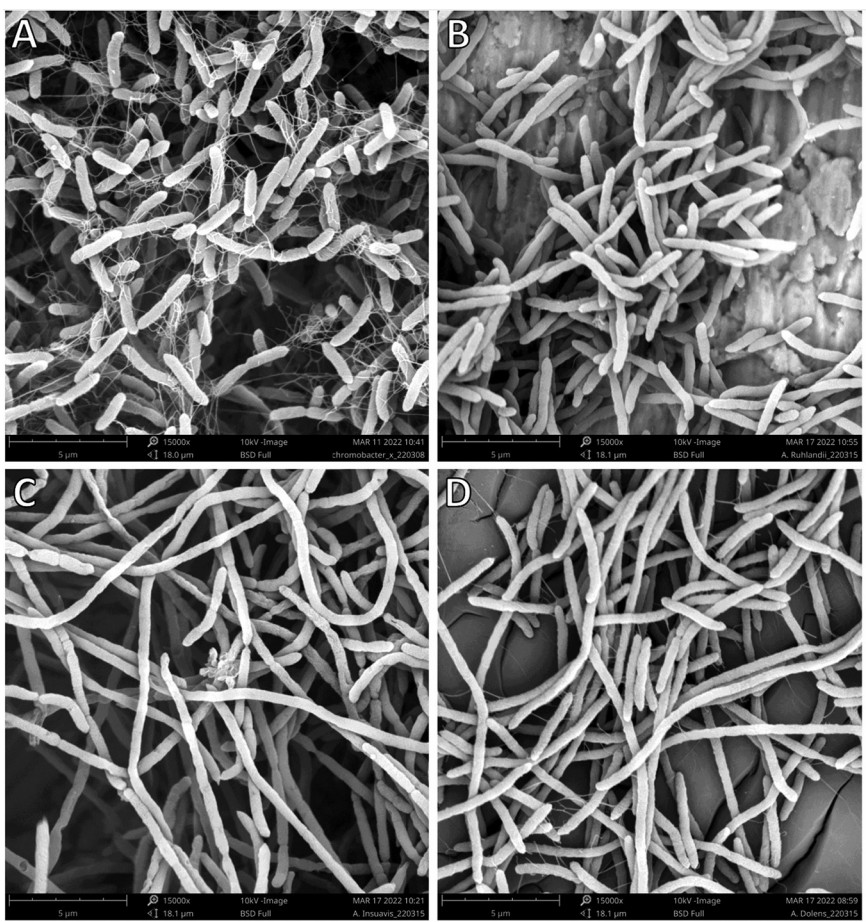

**FIG 2** *Achromobacter* type strains visualized using scanning electron microscopy at scale 5 $\mu$m. *A. xylosoxidans* (A) appear as short rods with abundant flagellae. *A. ruhlandii* (B) are shorter rods with very few visible flagellae. *A. insuavis* (C) have a filamentous appearance without visible flagellae. *A. dolens* (D) is filamentous similarly to *A. insuavis,* but with visible flagellae.

*P. aeruginosa* supernatants induced IL-6 secretion, but did not cause an IL-8 response from the airway epithelium. Compared with *P. aeruginosa*, exoproducts from all *Achromobacter* spp. except *A. insuavis* caused a significantly higher IL-6 secretion (Fig. 3A), and *A. xylosoxidans* and *A. dolens* exoproducts induced higher IL-8 secretion (Fig. 3B).

**Exoproducts from *Achromobacter* spp. activate neutrophils and monocytes.** The effects of bacterial exoproducts on neutrophil and monocyte activation were studied using peripheral blood from healthy donors. Whole blood was stimulated with bacterial supernatants, and activation of neutrophils and monocytes were subsequently measured as CD11b expression using flow cytometry. Supernatants from all *Achromobacter* species, as well as from *P. aeruginosa*, caused a significant increase in CD11b expression in both neutrophils and monocytes compared to medium control (Fig. 4A and B and Fig. S4A and B). *A. xylosoxidans*, *A. insuavis*, and *A. ruhlandii* caused a higher CD11b upregulation on neutrophils than *P. aeruginosa* (Fig. 4A). *A. xylosoxidans* and *A. insuavis* caused a higher CD11b upregulation on monocytes than *P. aeruginosa*, and *A. xylosoxidans* was also significantly more potent than *A. dolens*. (Fig. 4B).

Neutrophil activation by bacterial supernatants in whole blood was further confirmed by quantification of heparin-binding protein (HBP), which is a granular protein released from neutrophils upon activation. All species included in the study caused significant HBP release (Fig. S4C) compared to control, and all *Achromobacter* species caused a significantly higher release than *P. aeruginosa* (Fig. 4C).

**LPS is not the primary mediator of neutrophil and monocyte activation.** The exoproducts may contain LPS, which is a known proinflammatory agent present in the

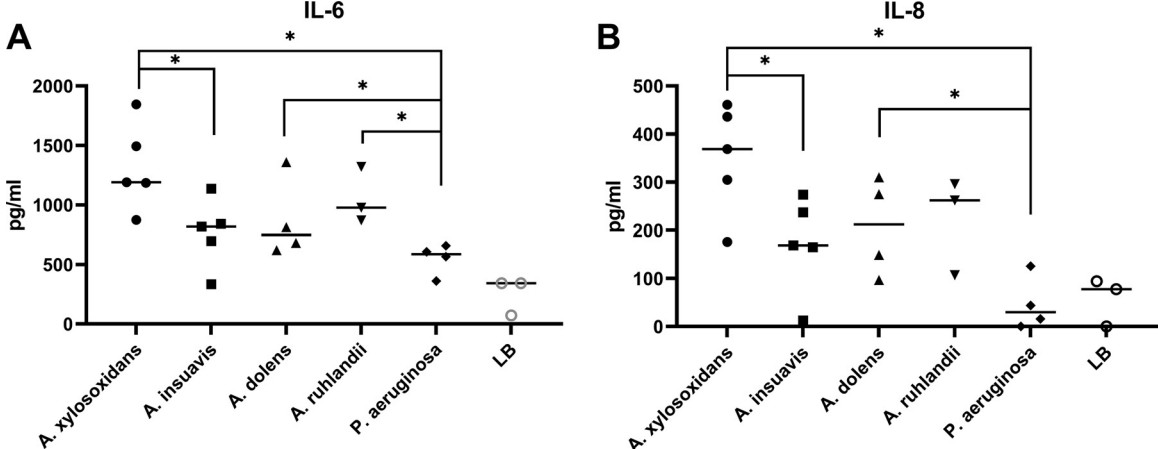

**FIG 3** Cytokine responses from CFBEo- cell cultures stimulated with 5% (vol/vol) bacterial supernatant. The Figure shows concentrations of IL-6 (A) and IL-8 (B) in the cell culture medium after 24h of stimulation. Each symbol represents a bacterial isolate and shows the mean value of three independent replicates. Bars illustrate the mean. Pairwise comparisons were made using Mann-Whitney U test (*, $P < 0.05$).

outer membrane of Gram-negative bacteria. In order to determine whether the observed proinflammatory properties of bacterial supernatants were caused by LPS, we stimulated whole blood with type strain supernatants that had been pre-incubated with or without polymyxin B. *P. aeruginosa* LPS was included as a control. Neutrophils were activated by all *Achromobacter* spp. supernatants but not by LPS (Fig. 5A). Monocytes were activated by LPS alone, and the stimulation caused by LPS could be inhibited by polymyxin B. In contrast, cell activation by *Achromobacter* supernatants was not inhibited by polymyxin B (Fig. 5B), suggesting that LPS is not the main proinflammatory mediator of the bacterial supernatants.

## DISCUSSION

In this study, we present evidence that *Achromobacter* exoproducts cause *in vitro* inflammation at similar or stronger levels than *P. aeruginosa*. None of the four *Achromobacter* species included in the study showed a consistent difference in inflammatory potential compared to the others. While *A. xylosoxidans* supernatants tended to induce the highest mean inflammatory responses, the differences were only statistically significant compared to *A. insuavis* with regard to interleukin induction, and to *A. dolens* in monocyte activation. Exoproducts from all four *Achromobacter* species activated neutrophils equally. Previous

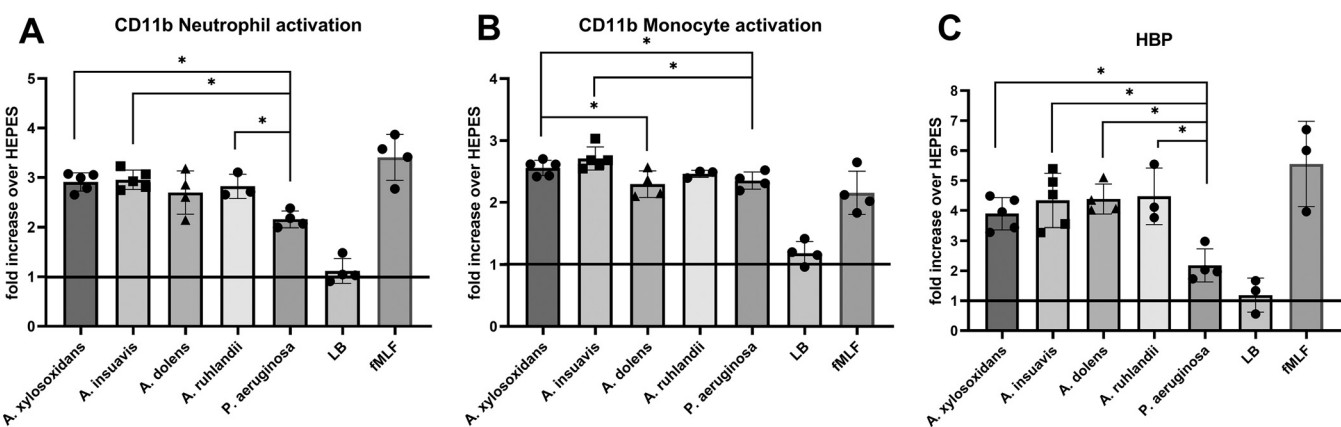

**FIG 4** Leukocyte activation by stimulation with 5% (vol/vol) bacterial supernatant in whole blood. CD11b expression (A, B) and HBP release (C) are calculated as fold increase over negative control (HEPES). The solid line represents no increase compared to negative control (1-fold). fMLF is used as positive control. All supernatants caused significant increase of CD11b expression in neutrophils (A) and monocytes (B) compared to LB. The supernatants of all species studied gave rise to a significant HBP release compared to LB (C). Each dot represents one bacterial isolate or control as the average value of 3 repeats. Bars represent the mean with SD. Pairwise comparisons were made using Mann-Whitney U test (*, $P < 0.05$).

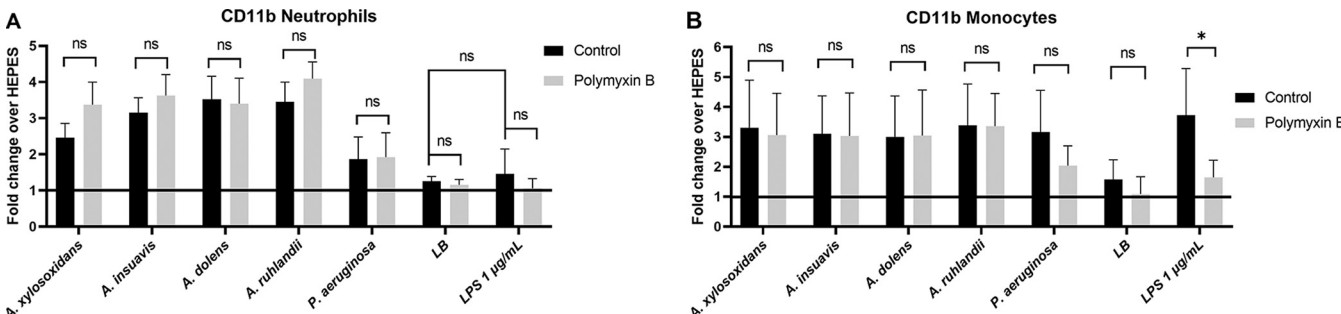

**FIG 5** Neutrophils (A) and monocytes (B) from whole blood were stimulated with type strain bacterial supernatants, LB medium or LPS in the absence or presence of 40 $\mu$g/mL polymyxin B. Activation was assessed by CD11b upregulation and analyzed with flow cytometry. Bars express the mean with SD of fold change over negative control (HEPES) in 3 repeats. Pairwise comparisons were made using Mann-Whitney U test (*, $P < 0.05$).

studies have shown that exoproducts from *A. xylosoxidans* are capable of mounting an inflammatory response from lung epithelium (23), but this study is to our knowledge the first to compare proinflammatory properties between different *Achromobacter* species.

At our clinical center, 29% of the isolates identified as *A. xylosoxidans* in the clinical routine diagnostics were in fact non-*xylosoxidans*, and similar results have been reported from other regions. The most common species, *A. xylosoxidans*, is expected to account for 40 to 60% of all *Achromobacter* infections in CF with geographical variance greatly affecting the prevalence. *A. ruhlandii* is the second most common species in the Americas (18, 24, 25), whereas *A. insuavis* and *A. dolens* are more common in Europe (19, 26). Improved species identification with whole-genome sequencing and updated MALDI-TOF-MS libraries (27) will be of importance for further studies, e.g., on the clinical outcomes of patients with non-*xylosoxidans* infections.

Interestingly, *P. aeruginosa* exoproducts induced a lower inflammatory response compared to many of the *Achromobacter* species in this study. *P. aeruginosa* is one of the most prevalent and well-studied CF pathogens. It is known to cause inflammatory responses leading to lung damage (28), and persistent infection increases the risk of deteriorating lung function (29, 30). Secreted factors from *P. aeruginosa* such as proteases and pyocyanins have been shown to have important functions in disrupting host epithelium and modulating inflammatory responses (31, 32). If immune responses to *Achromobacter* are even stronger than that generated to *P. aeruginosa*, it supports the hypothesis that *Achromobacter* plays an actively pathogenic role in CF lung disease. In support of our results, it has been shown that *Achromobacter* infections correlate to inflammatory markers in CF sputum equivalent to *P. aeruginosa* (33).

LPS from *Achromobacter* has been confirmed to have proinflammatory properties (34). However, LPS does not appear to be the main contributing factor to the inflammatory responses observed in this study, since neutralisation of LPS with polymyxin B did not affect the inflammatory potential of bacterial supernatants. We speculate that *Achromobacter* spp. release other mediators that induce cytokine release from the airway epithelium, including secretion of the strong neutrophil chemoattractant IL-8 (35). This aids neutrophil and monocyte recruitment to the lung, where *Achromobacter* exoproducts may further contribute to subsequent leukocyte activation. Continuing proteomic studies are needed to characterize the contents of the bacterial supernatants and their potentially immunogenic components, since the composition of the exoproducts were not addressed in this study. It is possible that the supernatant fraction contains not only secreted products, but also shed surface proteins or even cytosolic components that may have been released during preparation.

Despite the morphological differences observed in EM images, *Achromobacter* species did not differ in motility or biofilm capabilities. In agreement with other studies, *Achromobacter* biofilm formation differs between isolates and appears to vary more between isolates than between species (36). This is likely due to differences in host conditions and bacterial with-host evolution during long-term colonization of the CF

lung (37). Despite being generally less capable of forming biofilms than *P. aeruginosa*, *Achromobacter* spp. infections present a great clinical treatment challenge due to both innate and acquired antibiotic resistance mechanisms (38–40).

An important limitation of our study is the small number of isolates and the fact that they were obtained from both chronic and sporadic CF infections as well as from reference laboratories. The possible differences between clinical isolates from various types of infection adds to the complexity of interpreting these results. Additionally, there is a discrepancy between *in vitro* and *in vivo* growth conditions. The CF lung is a complex environment with various microbiological niches. Interactions with the immune system, other microorganisms, biofilm formation and antibiotic pressure may affect virulence *in vivo*. In addition, we studied secreted exoproducts and did not evaluate the effect of whole bacteria. Studies on inflammatory markers in sputum and lung function over time in patients chronically infected with different *Achromobacter* species would add valuable information. Persistent infections can give rise to in-host evolution affecting the cellular morphology, such as downregulation of flagellins (37, 41), compared to the type strains used in imaging. Further studies are necessary to confirm whether the morphological differences observed are consistent between different isolates of the same species or may be affected by long-term colonization and adaptation to the CF lung.

Another limitation is that peripheral blood cells were obtained from healthy donors and not CF patients, whose neutrophils and monocytes can be affected by the CFTR mutation even in the absence of infection (1). However, this strengthens the reproducibility of results as CF patients vary greatly in disease severity, ongoing infections and treatment regimens that may affect the leukocyte responsiveness.

In conclusion, our results point toward the possibility that *Achromobacter* spp. may play an actively pathogenic role in CF lung infections. Whereas the inflammatory responses to exoproducts did not consistently vary between the four investigated species, all *Achromobacter* spp. supernatants caused an equivalent or stronger response compared to *P. aeruginosa* and implies that treatment of both infections are important in the management of CF lung disease.

## MATERIALS AND METHODS

**Bacterial isolates, species identification, and culturing conditions.** The bacterial isolates included in this study are presented in Table 1. Clinical isolates of *Achromobacter* spp. and *P. aeruginosa* from CF sputum cultures were obtained from the department of Clinical Microbiology, Skåne University Hospital, Lund, Sweden. Species identification was done according to standard laboratory methods, including Matrix-Assisted Laser Desorption/Ionization Time Of Flight Mass Spectrometry (MALDI-TOF MS) (Bruker Daltonics, Bremen, Germany) (42). All clinical *Achromobacter* isolates had been identified as *A. xylosoxidans* according to MALDI-TOF. To further distinguish between *Achromobacter* species, sequencing of the *nrdA* locus was performed as described previously, using the primers P1: GAACTGGATTCCCGACCTGTTC, P2: TTCGATTTGACGTACAAGTTCTGG (43). Sequencing was performed by Eurofins Genomics, Ebersberg, Germany. Species identification based on *nrdA* sequences was performed using the *Achromobacter* typing database in PubMLST. Results from *nrdA* sequencing are presented in supplementary file 1.

Additional *Achromobacter* isolates, including type strains of each of the four *Achromobacter* species studied were purchased from the Culture Collection University of Gothenburg (CCUG), Sweden. The *P. aeruginosa* reference strain PAO1 was a kind gift from Arne Egesten, Lund University.

All strains were grown in Luria Bertani (LB) broth (Sigma-Aldrich L3022), shaking at 37°C, unless otherwise specified.

**Swimming motility.** To evaluate the bacterial swimming motility, a similar-sized colony of each isolate was spotted in the center of a 0.3% agar plate. The plates used to promote migration contained a low concentration of nutrients (2 g/ L $(NH_4)_2SO_4$, 6 g/ L $Na_2HPO_4$, 3 g/ L $KH_2PO_4$, 3 g/ L NaCl, 1 mM $MgCl_2$, 0.1 mM $CaCl_2$, 0.01 mM $FeCl_3$, 2.5 mg/ L thiamine, supplemented with 0.5% (wt/vol) glucose, and Casamino Acids (Merck)) (37). The plates were incubated at 37°C for 24 h, and the diameters of migration from the site of inoculation were measured.

**Biofilm formation assay.** Overnight cultures of each isolate were normalized to the same $OD_{620}$, diluted 1:10 in LB (18 $\mu$L + 162 $\mu$L), and grown for 72 h in a sterile 96-well plate covered with Breathe-Easier sealing film (Diversified Biotech). The plate was incubated at 37°C with 5% $CO_2$.

The plate was then washed three times with sterile phosphate-buffered saline (PBS) and fixated with 200 $\mu$L methanol for 10 min. The methanol was removed and the plate dried in room temperature for 1 h. 0,1% crystal violet was added in a volume of 160 $\mu$L and the plate was incubated for 4 min. The wells were then washed three times with PBS. Remaining stained biofilm was dissolved with 20:80 acetone-ethanol and diluted 1:4 before absorbance readout at 550 nm.

**Preparation of exoproducts.** Overnight cultures were adjusted to $OD_{620} = 0.650$ by dilution, followed by centrifugation for 10 min at 1500 g. $OD_{620} = 0.650$ was the lowest measurement of the overnight cultures, and all other samples were diluted to match this value in order to normalize the concentrations. The cell free supernatants were collected and filter sterilized using a 0.22 $\mu$m filter (Millex-GP Syringe Filter Unit, Merck Millipore). 10 $\mu$L of each supernatant was plated on LB agar overnight to ensure the absence of CFU. Three batches were prepared in order to use different culture supernatants for each experimental repeat. The supernatants were stored at $-20°C$ in aliquots. These preparations, here referred to as exoproducts, contains released bacterial products that could be both actively secreted or passively released from the surface. In downstream experiments, 5% (vol/vol) bacterial supernatant was used in cell stimulation assays based on titration pilot studies (Fig. S1).

**Cell culture.** The human CF bronchial epithelial cell line CFBEo- (product no. scc151, Millipore, Temecula, CA) was maintained in minimal essential medium (MEM) with Earle's salts and L-glutamine (Gibco 31095-029) containing 10% fetal bovine serum (FBS) (Gibco 11550356) and 2% 1× penicillin-streptomycin solution (Gibco 15140122). Cell culture plates and flasks were coated with 0.5 mg/mL human fibronectin (Sigma-Aldrich M2279), 1 mg/mL bovine serum albumin fraction V (Saveen o Werner AB, Sweden), and 3 mg/mL bovine collagen solution (PureCol, Advanced BioMatrix no. 5005) in MEM. For experiments, cells were seeded in a coated 24-well plate and grown until reaching approximately 90% confluence. Cells were then washed twice in warm PBS (Gibco 14190-144) and stimulated with 5% (vol/vol) sterile bacterial supernatant in cell medium. Sterile filtered LB and cell medium were used as negative controls. Cells were incubated with bacterial supernatants and controls for 24 h, after which they were visually inspected to assess viability. The medium was collected and stored at $-20°C$ until analysis.

**Cytokine ELISA.** Concentrations of interleukin (IL)-6 and IL-8 in the cell free medium were analyzed using DuoSet ELISA kits (DY206-05 and DY208-05, R&D Systems, Minneapolis, MN) according to the manufacturer's instructions. Cell supernatants were analyzed in 1:10 dilution for IL-6 and at 1:1 for IL-8. The cytokine response from cell medium alone was subtracted from the results, and sterile LB medium was used as negative control.

**Flow cytometry.** Blood collection from healthy donors was approved by the regional Ethical Review Board in Lund (reference number 2008/657). Blood was collected from 3 healthy donors into sodium citrate tubes. 50 $\mu$L was transferred to flow cytometry tubes, diluted in 45 $\mu$L 4-(2-hydroxyethyl)-1-piperazineethanesulfonic acid (HEPES) buffer (pH 7.4; 10 mM HEPES, 150 mM NaCl, 5 mM KCl, and 1 mM MgSO4), and stimulated with 5 $\mu$L bacterial supernatants or LB (5% vol/vol, final concentration) to a final sample volume of 100 $\mu$L. 1 $\mu$M N-Formylmethionyl-leucyl-phenylalanine (fMLF) (Sigma-Aldrich, F3506) was used as a positive control for leukocyte activation and HEPES was used as negative control. After incubation at room temperature for 2 h, samples were stained using 5 $\mu$L each of mouse-anti-human CD45-FITC (BD Pharmingen 561865 clone HI30) to enable gating of leukocyte populations, and CD11b-PE-Cy5 (BD Pharmingen 555389 clone ICRF44) as a marker of neutrophil activation (44, 45). Samples were incubated for 15 min at room temperature in the dark. Red blood cells were lysed and samples were fixed using 1-step Fix/Lyse solution (Invitrogen eBioscience 00–5333-57), and incubated in the dark at room temperature for 1 h. Samples were then centrifuged (500 g, 5 min), the supernatant discarded, and cells resuspended in 500 $\mu$L PBS. Samples were analyzed using a BD Accuri C6 flow cytometry during the same day. Isotype control experiments did not indicate any nonspecific binding effects of the antibodies.

**Heparin-binding protein (HBP) ELISA.** One huyndred $\mu$L citrated whole blood from healthy donors was diluted in 850 $\mu$L PBS and stimulated with 50 $\mu$L bacterial supernatants or LB (5% vol/vol, final concentration). 1 $\mu$M fMLF was used as positive control. Samples were incubated for 30 min at 37°C with gentle shaking. The samples were then centrifuged (300 g, 15 min) and the HBP contents of the supernatants were analyzed by ELISA according to manufacturer's instructions (Axis-Shield FMHBP100IUO). The experiment was repeated 3 times with separate donors and supernatant batches.

**Inhibition of LPS.** Supernatants of type strains of the four *Achromobacter* species were used for further analysis regarding LPS activity and compared to the supernatant of *P. aeruginosa* strain PAO1. Supernatants were preincubated with or without 40 $\mu$g/mL polymyxin B (Sigma-Aldrich) for 20 min at 37°C. *P. aeruginosa* LPS (Merck) was included as a control (1 $\mu$g/mL, final concentration), with or without polymyxin B. Peripheral blood was collected from 3 healthy donors and stimulated with bacterial supernatants, HEPES, LPS, or fMLF according to the flow cytometry protocol above.

**Electron microscopy.** The four *Achromobacter* type strains were grown to midlogarithmic phase with an $OD_{620}$ of 0.450 (*A. insuavis* $OD_{620} = 0.320$). Samples were centrifuged and resuspended to a bacterial concentration of approximately $2 \times 10^9$ CFU/mL in fixative (2,5%) glutaraldehyde (Merck) in 0.15 M sodium cacodylate pH 7.2 (Sigma-Aldrich), and incubated at room temperature overnight.

Samples were washed with 0.15 M sodium cacodylate and dehydrated with an ascending ethanol series from 50% (vol/vol) to absolute ethanol. The specimens were then subjected to critical-point drying with carbon dioxide and absolute ethanol was used as an intermediate solvent. The tissue samples were mounted on aluminum holders, sputtered with 20 nm palladium/gold, and examined in a correlative light and scanning electron microscope (DELPHI PhenomWorld, Delmic, Netherlands) at 10 kV (46).

**Statistical methods.** All statistical analysis and graphical presentations were performed using Prism 8 software (Graph-Pad Software, San Diego CA). Comparisons between groups were made using Mann-Whitney U tests. Two-tailed $P < 0.05$ was regarded as statistically significant.

**Ethics statement.** Blood collection from healthy donors was approved by the regional Ethical Review Board in Lund (reference number 2008/657). Written informed consent was obtained from all participants.

## SUPPLEMENTAL MATERIAL

Supplemental material is available online only.

**SUPPLEMENTAL FILE 1**, DOCX file, 0.02 MB.
**SUPPLEMENTAL FILE 2**, DOCX file, 0.03 MB.
**SUPPLEMENTAL FILE 3**, TIF file, 0.2 MB.
**SUPPLEMENTAL FILE 4**, TIF file, 3.8 MB.
**SUPPLEMENTAL FILE 5**, TIF file, 0.8 MB.
**SUPPLEMENTAL FILE 6**, TIF file, 1 MB.

## ACKNOWLEDGMENTS

We thank the EM unit, Infection Medicine, Lund University, for access to instruments and expertise in electron microscopy.

C.S. planned and performed all studies except electron microscopy and wrote the manuscript. M.B. performed the electron microscopy. O.S. took part in study design of flow cytometry experiments. L.I.P. planned, supervised, and funded all studies. All authors read and approved the manuscript.

This study was funded by the MIMS Clinical Research Fellows (grant 81226), the Royal Physiographic Society of Lund (grant F2020/1782), the Swedish Heart and Lung foundation (grant F2022/2227), the Alfred Österlund foundation (grant F2022/190), the Swedish Heart and Lung Association (grant F2022/620), the Swedish Cystic Fibrosis Association (grant F2022/625), and the Knut and Alice Wallenberg Foundation, the medical faculty at Lund University, and Region Skåne (grant 81234).

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
