## [Reviewer comments · Microbiology Spectrum]

Microbiology Spectrum

Exoproducts of the most common *Achromobacter* species in cystic fibrosis evoke similar inflammatory responses *in vitro*

Cecilia Sahl, Maria Baumgarten, Oonagh Shannon, and Lisa Pålman

Corresponding Author(s): Lisa Pålman, Lunds Universitet

Review Timeline:

Submission Date:	January 16, 2023
Editorial Decision:	March 7, 2023
Revision Received:	May 5, 2023
Accepted:	May 16, 2023

Editor: Susan Realegeno

Reviewer(s): Disclosure of reviewer identity is with reference to reviewer comments included in decision letter(s). The following individuals involved in review of your submission have agreed to reveal their identity: Sura Ali alasadi (Reviewer #2); Helene MARCHANDIN (Reviewer #3)

Transaction Report:

DOI: <https://doi.org/10.1128/spectrum.00195-23>

March 7, 2023

Dr. Lisa I Pålman
Lunds Universitet
Department of Clinical Sciences Lund, Division of Infection Medicine
BMC B14, Baravägen 27
Lund SE-223 63
Sweden

Re: Spectrum00195-23 (Exoproducts of the most common *Achromobacter* species in cystic fibrosis evoke similar inflammatory responses *in vitro*)

Dear Dr. Lisa I Pålman:

Link Not Available

Sincerely,

Susan Realegeno

Journals Department
Reviewer comments:

Reviewer #1 (Comments for the Author):

Exoproducts of the most common *Achromobacter* species in cystic fibrosis evoke similar inflammatory responses *in vitro*. Sahl et al.

In this manuscript by Sahl et al different *Achromobacter* species are tested for their ability to induce an inflammatory response. Different species were used as all are clinically relevant for patients with cystic fibrosis (CF). Importantly, the current diagnostic tests are insufficient to differentiate between *Achromobacter* sp. without a nucleotide sequence analysis. Thus, the first step was

to verify the identification of the species, followed by phenotypic characterization and then testing for the induction of inflammatory cytokines in vitro. *Pseudomonas aeruginosa* was used as a positive control as this bacterium is a well-known and well-characterized chronic colonizer of CF patients. The results indicate that there is wide variation in the phenotypic characteristics of different species and even different strains within a confined species. Filtered supernatants from *Achromobacter* species induced the production of IL-6 and IL-8 from human CF epithelial cells to levels that exceeded the same material from *P. aeruginosa*. The same result occurred when normal human monocytes and neutrophils were analyzed for activation. Polymixin B did not seem to impact the activation potential of the supernatants, suggesting that the common antigen between all strains, LPS, is unlikely to be the main inflammatory inducing factor. This is a focused, well-executed study demonstrating that several species of *Achromobacter*, routinely isolated from CF patients, induce inflammatory mediators and activate monocytes and neutrophils to levels that exceed that of *P. aeruginosa*. The discussion of the results is thoughtful and does not overreach the data. Importantly, the authors discuss the limitations of the current study. A few minor points should be addressed.

Comments for the Author's consideration

1. The number of replicate experiments is reported for the cytokine and cellular activation studies, however, it is unclear how many times the assays were repeated for bacterial phenotypic characterization. If only one experiment was done, the variation in the outcomes may be due to random chance, growth conditions, amount of bacterial lysis or just the wrong growth conditions for the wrong species. Perhaps this is a limitation of the study that should be included in the discussion.
2. The exoproduct designation can be misinterpreted. In a sense, it is true that the inflammatory inducing product might be secreted into the supernatant as the word exoproduct implies. It could also be true that it is a cellular component that is shed during growth.
3. Figure 5 refers to a "type" strain used in this analysis. Does this refer to the first strain listed in Table 1 for each species?
4. Under the heading Cell culture conditions, the authors state that the cells were incubated with a dilution of the bacterial supernatant, incubated for 24 h, and then visually inspected to assess viability. It might be interesting to measure viability to be able to determine whether toxic activity in the supernatant is partially responsible for inducing cytokine production and or activation of monocytes and neutrophils. If most the cells seemed to be in good shape then stating that would be important.
5. A 5% solution is a relatively low amount to induce these activities, suggesting that whatever is made is quite potent. It would be interesting to identify the component as potentially common between a variety of species, but this is clearly beyond the scope of the current manuscript. It might be interesting to standardize the supernatants and then do a titration to determine, whether there are production differences.

Reviewer #2 (Comments for the Author):

- 1-In line 15 its better to change "inflammation" into invitro inflammatory response.
- 2-When you mention a scientific term for the first time in the paper you should write it as a full scientific term. Lines(81,145,150,167,168) .
- 3-The reference that you depended on for the (MALDI-TOF-MS)analysis for bacterial identification is updated by the following reference , T Garrigos , T. , et al., Development of a database for the rapid and accurate routine identification of *Achromobacter* species by matrix-assisted laser desorption/ionization-time-of-flight mass spectrometry (MALDI-TOF MS).*J Clin Microbiol Infect*, 2021.27(1):p. 126.e1-126.e5 which may explain the differences in results between the two methods of bacterial identifications techniques you have been used .
- 4-Almost all the methods listed in this paper is not supported by references such as swimming motility, biofilm formation assay, preparation of exoproducts, cell culture and electron microscopy.
- 5-In line 136, explain the reason for using the concentration OD620=0.650.
- 6-Where is the results of *nrdA* sequencing alignment?
- 7- Please rephrase the sentence in line 248.
- 8- In line 251 you wrote that *Achromobacter* species except *A. insuavis* induced a significant IL-6 and IL-8 response from CF lung epithelium ,while you neglect the fact that 4 out of 5 *A. insuavis* strains where inducible for the tested interleukins.
- 9-In line 252 *A. xylooxidans* caused significantly higher responses than all other tested *A. spp.* Strains.
- 10- In line 255 change (supplementary Figure 1A) into Figure 3-A.
- 11- In line 257 change (supplementary Fig. 1B)into Figure 3-B.
- 12-At the end of line 265 , please rephrase the sentence.
- 13- Explain the role of measuring CD11b expression in this study.
- 14- In line 272 change (supplementary Fig. 2A) into Figure.4-A.
- 15- In line 274 change (supplementary Fig. 2B) into Figure.4-B, and you missed writing about *A. ruhlandii* results.

16- In line 279 change (supplementary Fig. 2C) into Figure.4-C.

17-There are some differences in results between Figure 4A and Figure 5A in respect to *A. xylosoxidans* with *A. dolens*, although they were tested about the same CD11b neutrophils.

18-In line 321 there is no reference to approve the statement.

19-In line 352you wrote about antibiotic resistance but you haven't studied this.

Reviewer #3 (Comments for the Author):

Sahl et al. reported interesting results on the ability of members of 4 different species of *Achromobacter* to induce inflammation. *Achromobacter* spp. are considered as emerging pathogens in CF and there is still a need for deeper knowledge on individual species within the genus.

Strengths of the paper are that:

- Strains were identified using *nrdA* gene sequencing (but see major comment below).

- Production of IL-6 and IL-8 was measured, activation of neutrophils and monocytes as well of the role of LPS in this activation were evaluated for each of these 4 species and in comparison with *Pseudomonas aeruginosa*, the major and most studied pathogen in Cystic Fibrosis.

Thus the study provides original results on in vitro inflammatory responses of 4 species in the genus *Achromobacter*, species other than *A. xylosoxidans* being very rarely studied.

The authors also propose a study of biofilm formation and swimming motility for the clinical strains included in the study and a morphological study using scanning electron microscopy (SEM) of the type strains of the 4 *Achromobacter* species.

These sections of the manuscript, although interesting, did not add to the core of the subject of this manuscript. Proof of this is that they are not mentioned in the title and they did not appear in the abstract. The reader is thus surprised to find such data when reading the manuscript. I strongly recommend these parts to be withdrawn from this manuscript for more clarity (so I did not add any additional comment on these sections).

Major comments:

- Missing important information related to the isolates, principally were they recovered from sporadic, intermittent or chronic infections as this may affect the strains' characteristics due to adaptation over the colonization?

- Missing important information on how *nrdA* gene sequences were analyzed: length of the sequence considered in the analysis? BLAST? PubMLST? Phylogenetic analysis (recommended)?

Materials and Methods and also Results (lines 214-216)

Very important for this study is the correct assignment to a species within the genus *Achromobacter*.

- From my point of view comparative results with *P. aeruginosa* are as important than results obtained compared to the negative control. I recommend supplementary figures to be incorporated in the manuscript (in place of deleted parts on biofilm, motility, SEM).

- Despite the few number of strains in each species, variability of the results has to be considered, described and discussed. This intraspecific variability may be related to patient colonization type.

- Lines 324-326: I do not agree with the authors. The precise identification of the *Achromobacter* species is still required to increase knowledge on each of these species and this cannot be ascertain after studying 4 clinical isolates per species.

Other comments:

Please note that there should be a point after spp. (correct throughout the entire manuscript including legends to figures) (and spp. not in italics - line 293).

Abstract

Line 36: replace found by identified.

Lines 39-40: identification method has to appear here.

Line 42: *Pseudomonas* instead of *P.* here.

Lines 45-52: for more clarity, it would be preferable to indicate that the species did not differ in causing inflammatory responses despite the absence of IL-6 & IL-8 production shown for *A. insuavis*.

Not a single word on biofilm, motility and SEM.

Introduction

Line 77: missing word? of environmental origin.

Lines 80-83: please underline the differences between studies that may have accounted for the distinct results.

Line 86, 87, 91: spp. should be replaced by species.

Line 88: classified to be replaced by misidentified.

Lines 88-89: OK for the oldest publications (to be specified and also specify which methods you refer to).

Line 89: or not in italics.

Line 95: characterized.

Materials and methods

Lines 99-100: Indicate here number of strains.

Line 106: nrdA gene sequencing: incomplete methodology: analysis of the sequences is a very important point to be sure of the species assignment reported (see major comment and comments on Results).

Line 138: were instead of was.

Results

A robust assignment to species is required, based on nrdA-based phylogeny (as in the PubMLST database, there are some wrong assignments, and as a percentage of nrdA gene sequence similarity above the threshold published by Spilker et al. in 2012 is not be able to distinguish between strains of the 3 closely related species: *A. xylosoxidans*, *A. ruhlandii* and *A. dolens*. A figure - nrdA gene-based phylogenetic tree - would be welcome in this type of paper in which a precise identification of the strains is mandatory.

Lines 221-226 to be deleted. In addition to the fact that this is not strictly related to the research subject, for these analyzes, the very low number of strains in each species is an important limitation. Mix of clinical and reference strains is also probably not appropriate due to distinct characteristics usually reported, particularly in the CF context.

Discussion

Lines 315-318: Reference 35 to be cited here?

Line 338-341: these are surprising results that should be deeper discussed. Is there any other Gram-negative bacilli displaying this specific feature in the literature?

Lines 347-354: to be deleted.

Line 355: add to the limitations of the study the very low number of strains studied for each species.

Staff Comments:

Preparing Revision Guidelines

Please return the manuscript within 60 days; if you cannot complete the modification within this time period, please contact me. If you do not wish to modify the manuscript and prefer to submit it to another journal, please notify me of your decision immediately so that the manuscript may be formally withdrawn from consideration by Microbiology Spectrum.

Corresponding authors may join or renew ASM membership to obtain discounts on publication fees. Need to upgrade your

membership level? Please contact Customer Service at Service@asmusa.org.

- 1-In line 15 its better to change “inflammation” into invitro inflammatory response.
- 2-When you mention a scientific term for the first time in the paper you should write it as a full scientific term. Lines(81,145,150,167,168) .
- 3-The reference that you depended on for the (MALDI-TOF-MS)analysis for bacterial identification is updated by the following reference , T Garrigos , T. , et al., Development of a database for the rapid and accurate routine identification of Achromobacter species by matrix-assisted laser desorption/ionization-time-of-flight mass spectrometry (MALDI-TOF MS).J Clin Microbiol Infect, 2021.27(1):p. 126.e1-126.e5
which may explain the differences in results between the two methods of bacterial identifications techniques you have been used .
- 4-Almost all the methods listed in this paper is not supported by references such as swimming motility, biofilm formation assay, preparation of exoproducts, cell culture and electron microscopy.
- 5-In line 136, explain the reason for using the concentration $OD_{620}=0.650$.
- 6-Where is the results of *nrdA* sequencing alignment?
- 7- Please rephrase the sentence in line 248.
- 8- In line 251 you wrote that *Achromobacter* species except *A. insuavis* induced a significant IL-6 and IL-8 response from CF lung epithelium ,while you neglect the fact that 4 out of 5 *A. insuavis* strains where inducible for the tested interleukins.
- 9-In line 252 *A. xylosoxidans* caused significantly higher responses than all other tested *A. spp.* Strains.
- 10- In line 255 change (supplementary Figure 1A) into Figure 3-A.
- 11- In line 257 change (supplementary Fig. 1B)into Figure 3-B.
- 12-At the end of line 265 , please rephrase the sentence.
- 13- Explain the role of measuring CD11b expression in this study.
- 14- In line 272 change (supplementary Fig. 2A) into Figure.4-A.
- 15- In line 274 change (supplementary Fig. 2B) into Figure.4-B, and you missed writing about *A. ruhlandii* results.
- 16- In line 279 change (supplementary Fig. 2C) into Figure.4-C.
- 17-There are some differences in results between Figure 4A and Figure 5A in respect to *A. xylosoxidans* with *A. dolens*, although they were tested about the same CD11b neutrophiles.
- 18-In line 321 there is no reference to approve the statement.
- 19-In line 352you wrote about antibiotic resistance but you haven’t studied this.

- 1-In line 15 its better to change “inflammation” into invitro inflammatory response.
- 2-When you mention a scientific term for the first time in the paper you should write it as a full scientific term. Lines(81,145,150,167,168) .
- 3-The reference that you depended on for the (MALDI-TOF-MS)analysis for bacterial identification is updated by the following reference , T Garrigos , T. , et al., Development of a database for the rapid and accurate routine identification of Achromobacter species by matrix-assisted laser desorption/ionization-time-of-flight mass spectrometry (MALDI-TOF MS).J Clin Microbiol Infect, 2021.27(1):p. 126.e1-126.e5
which may explain the differences in results between the two methods of bacterial identifications techniques you have been used .
- 4-Almost all the methods listed in this paper is not supported by references such as swimming motility, biofilm formation assay, preparation of exoproducts, cell culture and electron microscopy.
- 5-In line 136, explain the reason for using the concentration $OD_{620}=0.650$.
- 6-Where is the results of *nrdA* sequencing alignment?
- 7- Please rephrase the sentence in line 248.
- 8- In line 251 you wrote that *Achromobacter* species except *A. insuavis* induced a significant IL-6 and IL-8 response from CF lung epithelium ,while you neglect the fact that 4 out of 5 *A. insuavis* strains where inducible for the tested interleukins.
- 9-In line 252 *A. xylosoxidans* caused significantly higher responses than all other tested *A. spp.* Strains.
- 10- In line 255 change (supplementary Figure 1A) into Figure 3-A.
- 11- In line 257 change (supplementary Fig. 1B)into Figure 3-B.
- 12-At the end of line 265 , please rephrase the sentence.
- 13- Explain the role of measuring CD11b expression in this study.
- 14- In line 272 change (supplementary Fig. 2A) into Figure.4-A.
- 15- In line 274 change (supplementary Fig. 2B) into Figure.4-B, and you missed writing about *A. ruhlandii* results.
- 16- In line 279 change (supplementary Fig. 2C) into Figure.4-C.
- 17-There are some differences in results between Figure 4A and Figure 5A in respect to *A. xylosoxidans* with *A. dolens*, although they were tested about the same CD11b neutrophiles.
- 18-In line 321 there is no reference to approve the statement.
- 19-In line 352you wrote about antibiotic resistance but you haven’t studied this.

Responses to reviewers

Reviewer #1 (Comments for the Author):

1. The number of replicate experiments is reported for the cytokine and cellular activation studies, however, it is unclear how many times the assays were repeated for bacterial phenotypic characterization. If only one experiment was done, the variation in the outcomes may be due to random chance, growth conditions, amount of bacterial lysis or just the wrong growth conditions for the wrong species. Perhaps this is a limitation of the study that should be included in the discussion.

Swimming motility and biofilm formation assays were performed three times. The figure legend for these results has been updated to better clarify this.

2. The exoproduct designation can be misinterpreted. In a sense, it is true that the inflammatory inducing product might be secreted into the supernatant as the word exoproduct implies. It could also be true that it is a cellular component that is shed during growth.

This is a good point, and we specifically avoided the term 'secreted factors' for this reason. It is true that the bacterial supernatants may contain both secreted factors, shed surface proteins, and even cytosolic components. We have clarified this in both methods (lines 152-154) and discussion (lines 338-340).

3. Figure 5 refers to a "type" strain used in this analysis. Does this refer to the first strain listed in Table 1 for each species?

Yes, the type strain refers to representative strains obtained from the Culture Collection University of Gothenburg (CCUG) as described in Table 1. We have clarified in the text that we have included both clinical isolates and type strains on which the description of the species is based in the study (lines 238-239)

4. Under the heading Cell culture conditions, the authors state that the cells were incubated with a dilution of the bacterial supernatant, incubated for 24 h, and then visually inspected to assess viability. It might be interesting to measure viability to be able to determine whether toxic activity in the supernatant is partially responsible for inducing cytokine production and or activation of monocytes and neutrophils. If most the cells seemed to be in good shape then stating that would be important.

Upon visual inspection, there was no observable difference between wells exposed to *Achromobacter* supernatants, *Pseudomonas* supernatant or negative controls. Representative images have been added as a supplement (new supplementary figure 2) and the information has been included in the results section (lines 258-260). Cell viability assays using MTT and LDH were attempted but were not considered reliable since we observed highly variable results between repeats, likely due to interference of serum required in the cell medium.

5. A 5% solution is a relatively low amount to induce these activities, suggesting that whatever is made is quite potent. It would be interesting to identify the component as potentially common between a variety of species, but this is clearly beyond the scope of the current manuscript. It might be interesting to standardize the supernatants and then do a titration to determine, whether there are production differences.

The concentration of 5% was selected after a titration pilot study, where 5% was found to produce consistent responses in all studied cell types while not producing any background effect from the addition of bacterial growth medium. These results have been added to the supplement (new supplementary figure 1) and described in methods (lines 154-156). As described in methods, all supernatants were standardised with regards to each other (lines 146-148), by diluting every overnight culture to match the one with the lowest OD₆₂₀ (0.650).

Reviewer #2 (Comments for the Author):

1-In line 15 its better to change "inflammation" into invitro inflammatory response.

Edited

2-When you mention a scientific term for the first time in the paper you should write it as a full scientific term. Lines (81,145,150,167,168).

Edited

3-The reference that you depended on for the (MALDI-TOF-MS)analysis for bacterial identification is updated by the following reference , T Garrigos , T. , et al., Development of a database for the rapid and accurate routine identification of Achromobacter species by matrix-assisted laser desorption/ionization-time-of-flight mass spectrometry (MALDI-TOF MS).J Clin Microbiol Infect, 2021.27(1):p. 126.e1-126.e5 which may explain the differences in results between the two methods of bacterial identifications techniques you have been used.

While newer MALDI-TOF-MS databases are available, all *Achromobacter* isolates in this study were identified at the Clinical microbiology laboratory as *A. xylosoxidans* prior to 2021 when the new database was published. All diagnostic laboratories have not yet updated MALDI-TOF-MS databases for *Achromobacter*, and we hope that our study provides some support for the importance of up-to-date libraries in order to collect more data for e.g. retrospective studies on patient outcomes after colonisation with different species. The discussion section has been updated to clarify this (line 315-317).

4-Almost all the methods listed in this paper is not supported by references such as swimming motility, biofilm formation assay, preparation of exoproducts, cell culture and electron microscopy.

The reference for previous swimming motility assays from our lab has been added. Cell culture was performed according to manufacturer's instructions, the product code has been added to make it easier to find this. The biofilm formation assay is a simple crystal violet stain which has been described in detail in the methods section, and the preparation of exoproducts has also been described step by step.

The reference for electron microscopy has been added.

5-In line 136, explain the reason for using the concentration OD₆₂₀=0.650.

The clarification has been added (lines 146-148).

6-Where is the results of nrdA sequencing alignment?

The results have been added to supplementary file 1.

7- Please rephrase the sentence in line 248.

Edited

8- In line 251 you wrote that *Achromobacter* species except *A. insuavis* induced a significant IL-6 and IL-8 response from CF lung epithelium ,while you neglect the fact that 4 out of 5 *A. insuavis* strains were inducible for the tested interleukins.

Using statistical methods, the interleukin release induced by *A. insuavis* was not significantly higher than that of LB medium. However, due to the limited number of isolates included in this study, we do not want to speculate whether the difference is due to single outlier isolates or not. For the same reason, we refrain from drawing any conclusions that *A. insuavis* is less capable of causing inflammation in general than the other species.

9-In line 252 *A. xylosoxidans* caused significantly higher responses than all other tested *A. spp.* strains.

A. xylosoxidans gave rise to the highest mean response of all tested species, but this was only statistically significant when compared to *A. insuavis* and *P. aeruginosa*. The text has been updated to reflect this (lines 262-264).

10- In line 255 change (supplementary Figure 1A) into Figure 3-A.

Edited

11- In line 257 change (supplementary Fig. 1B)into Figure 3-B.

Edited

12-At the end of line 265 , please rephrase the sentence.

Edited

13- Explain the role of measuring CD11b expression in this study.

CD11b is a common marker of neutrophil and monocyte activation. We have included new references in the methods section.

14- In line 272 change (supplementary Fig. 2A) into Figure.4-A.

Edited

15- In line 274 change (supplementary Fig. 2B) into Figure.4-B, and you missed writing about *A. ruhlandii* results.

Figure numbering edited. *A. ruhlandii* did not induce more monocyte activation compared to *P. aeruginosa*, and we have therefore not commented on *A. ruhlandii* results.

16- In line 279 change (supplementary Fig. 2C) into Figure.4-C.

Edited

17-There are some differences in results between Figure 4A and Figure 5A in respect to *A. xylosoxidans* with *A. dolens*, although they were tested about the same CD11b neutrophiles.

In figure 5 A-B we did not include all 21 supernatants to test responsiveness to polymyxin B, but a representative type strain from each species in three repeats. Since we are not comparing groups but only the type strain, some differences can be expected between figure 4 and 5. We have clarified that only type strains were included in the experiment in the results section, line 291.

18-In line 321 there is no reference to approve the statement.

This line has been edited out in the current version.

19-In line 352 you wrote about antibiotic resistance but you haven't studied this.

The antibiotic resistances of *Achromobacter* have not been studied in this paper with regards to differences between species, but the resistance mechanisms of *Achromobacter* as a genus is known from previous studies as referenced to.

Reviewer #3 (Comments for the Author):

The authors also propose a study of biofilm formation and swimming motility for the clinical strains included in the study and a morphological study using scanning electron microscopy (SEM) of the type strains of the 4 *Achromobacter* species.

These sections of the manuscript, although interesting, did not add to the core of the subject of this manuscript. Proof of this is that they are not mentioned in the title and they did not appear in the abstract. The reader is thus surprised to find such data when reading the manuscript. I strongly recommend these parts to be withdrawn from this manuscript for more clarity (so I did not add any additional comment on these sections).

We believe that phenotypic characterisation of the bacterial isolates adds valuable information about differences and similarities between *Achromobacter* spp, and we have decided to retain this part in the manuscript. However, we acknowledge the reviewer's point of view, and we have included information about these results in the abstract (lines 40 and 46-48).

- Missing important information related to the isolates, principally were they recovered from sporadic, intermittent or chronic infections as this may affect the strains' characteristics due to adaptation over the colonization?

We agree that the type of infection is important information and it is now added to table 1. The *Pseudomonas* isolates included in the study were received anonymised from the clinical microbiology lab and we therefore only know that they were cultured from CF patients. We also lack information about the clinical isolates purchased from CCUG.

- Missing important information on how *nrdA* gene sequences were analyzed: length of the sequence considered in the analysis? BLAST? PubMLST? Phylogenetic analysis (recommended)? Materials and Methods and also Results (lines 214-216)

Very important for this study is the correct assignation to a species within the genus *Achromobacter*.

The results have been added to supplementary file 1, and described in methods (line 113-115).

- From my point of view comparative results with *P. aeruginosa* are as important than results obtained compared to the negative control. I recommend supplementary figures to be incorporated in the manuscript (in place of deleted parts on biofilm, motility, SEM).

We fully agree that the different inflammatory responses to *Achromobacter* and *Pseudomonas* exoproducts are highly interesting. For that reason, we have changed the figures so that the differences between *Achromobacter* spp and *Pseudomonas* (former supplementary figure) is now main figure 3A and B, and the original Figure 3 is now a supplement. We prefer not to include all figures in the manuscript, since they are presenting the same data but with different bars for statistical comparisons.

- Despite the few number of strains in each species, variability of the results has to be considered, described and discussed. This intraspecific variability may be related to patient colonization type.

Table 1 describing included isolates has been updated to include the colonisation types of the clinical isolates. Due to the small sample size, it is not possible to do sub-group analyses on sporadic/chronic infection. This has been added to the discussion as a limitation (lines 349-352).

- Lines 324-326: I do not agree with the authors. The precise identification of the *Achromobacter* species is still required to increase knowledge on each of these species and this cannot be ascertain after studying 4 clinical isolates per species.

We agree with the reviewer and we have rephrased the discussion accordingly. While we could not find any consistent differences between *Achromobacter* species in this study, correct identification is still important in order to gain more knowledge on the prevalence and outcomes of infections with different species, and we would not want the study to be used in support of forgoing this.

Other comments:

Please note that there should be a point after spp. (correct throughout the entire manuscript including legends to figures) (and spp. not in italics - line 293).

Edited

Abstract

Line 36: replace found by identified.

Edited

Lines 39-40: identification method has to appear here.

Edited

Line 42: *Pseudomonas* instead of P. here.

Edited

Lines 45-52: for more clarity, it would be preferable to indicate that the species did not differ in causing inflammatory responses despite the absence of IL-6 & IL-8 production shown for *A. insuavis*. Not a single word on biofilm, motility and SEM.

While the IL-6 and IL-8 responses were similar between the species, it would not be fully correct to state that they did not differ when *A. xylooxidans* responses were higher than *A. insuavis*. The results on biofilm, motility and SEM have been updated into the abstract.

Introduction

Line 77: missing word? of environmental origin.

Edited

Lines 80-83: please underline the differences between studies that may have accounted for the distinct results.

The parameters of “disease progression” can be studied in different ways, and with different selections of study endpoints it is not unsurprising for different studies to find an increase in hospitalisations but others not observing a decrease in FEV₁. In addition, the number of patients with *Achromobacter* infections is relatively small and has geographical variations in prevalence. This has been clarified in the background section (lines 81-83).

Line 86, 87, 91: spp. should be replaced by species.

Edited

Line 88: classified to be replaced by misidentified.

Edited

Lines 88-89: OK for the oldest publications (to be specified and also specify which methods you refer to).

Updated to specify that this refers to older methods.

Line 89: or not in italics.

Edited

Line 95: characterized.

Edited.

Materials and methods

Lines 99-100: Indicate here number of strains.

Edited

Line 106: nrdA gene sequencing: incomplete methodology: analysis of the sequences is a very important point to be sure of the species assignment reported (see major comment and comments on Results).

Edited, see above

Line 138: were instead of was.

Edited

Results

A robust assignment to species is required, based on nrdA-based phylogeny (as in the PubMLST database, there are some wrong assignments, and as a percentage of nrdA gene sequence similarity above the threshold published by Spilker et al. in 2012 is not be able to distinguish between strains of the 3 closely related species: *A. xylosoxidans*, *A. ruhlandii* and *A. dolens*).

A figure - *nrdA* gene-based phylogenetic tree - would be welcome in this type of paper in which a precise identification of the strains is mandatory.

The obtained sequences from *nrdA* sequencing have been provided in supplementary file 1, and the method section has been updated with the method for species assignation (lines 113-115).

Lines 221-226 to be deleted. In addition to the fact that this is not strictly related to the research subject, for these analyzes, the very low number of strains in each species is an important limitation. Mix of clinical and reference strains is also probably not appropriate due to distinct characteristics usually reported, particularly in the CF context.

While we would like to avoid deleting results, we have updated the discussion to clarify that the intra-species variation and mix of clinical and reference strains is an important limitation of these assays (lines 349-352).

Discussion

Lines 315-318: Reference 35 to be cited here?

Reference 35 is not discussing the prevalence of non-*xylosoxidans* infections, and references for this have been provided (19, 30-33)

Line 338-341: these are surprising results that should be deeper discussed. Is there any other Gram-negative bacilli displaying this specific feature in the literature?

Yes, it is a surprising result. We haven't ruled out that LPS plays a role in the observed cytokine response. Rather, as we mention in the discussion, we don't find support for LPS being the main driver of the inflammatory response in our experimental set-up.

Lines 347-354: to be deleted.

We have decided to retain this section of the discussion since we don't remove data from the manuscript, see also response to the first comment from reviewer 3.

Line 355: add to the limitations of the study the very low number of strains studied for each species.

Edited

May 16, 2023

Dr. Lisa I Pålman
Lunds Universitet
Department of Clinical Sciences Lund, Division of Infection Medicine
BMC B14, Baravägen 27
Lund SE-223 63
Sweden

Re: Spectrum00195-23R1 (Exoproducts of the most common *Achromobacter* species in cystic fibrosis evoke similar inflammatory responses *in vitro*)

Dear Dr. Lisa I Pålman:

Your manuscript has been accepted, and I am forwarding it to the ASM Journals Department for publication. You will be notified when your proofs are ready to be viewed.

Sincerely,

Susan Realegeno
Editor, Microbiology Spectrum
